# PROTAC-Mediated GSPT1 Degradation Impairs the Expression of Fusion Genes in Acute Myeloid Leukemia

**DOI:** 10.3390/cancers17020211

**Published:** 2025-01-10

**Authors:** Alicia Perzolli, Christian Steinebach, Jan Krönke, Michael Gütschow, C. Michel Zwaan, Farnaz Barneh, Olaf Heidenreich

**Affiliations:** 1Princess Maxima Center for Pediatric Oncology, 3584 CS Utrecht, The Netherlands; a.perzolli@prinsesmaximacentrum.nl (A.P.); c.m.zwaan-2@prinsesmaximacentrum.nl (C.M.Z.); f.barneh@prinsesmaximacentrum.nl (F.B.); 2Department of Pediatric Oncology, Erasmus MC/Sophia Children’s Hospital, 3015 GD Rotterdam, The Netherlands; 3Department of Pharmaceutical & Medicinal Chemistry, Pharmaceutical Institute, University of Bonn, An der Immenburg 4, 53121 Bonn, Germany; c.steinebach@uni-bonn.de (C.S.); guetschow@uni-bonn.de (M.G.); 4Department of Internal Medicine III, University Hospital Ulm, Albert-Einstein-Allee 23, 89081 Ulm, Germany; jan.kroenke@charite.de; 5Wolfson Childhood Cancer Research Centre, Newcastle University, Newcastle upon Tyne NE1 7RU, UK; 6Department of Hematology, University of Medical Center Utrecht, 3584 CX Utrecht, The Netherlands

**Keywords:** acute myeloid leukemia, CDK6, GSPT1, RUNX1::RUNX1T1, FUS::ERG, PROTACs, target therapy

## Abstract

Proteolysis-targeting chimeras (PROTACs) are an innovative and promising technology designed to degrade harmful proteins associated with cancer. While these drugs have demonstrated significant potential in treating various cancer types, their application in pediatric acute myeloid leukemia (AML) remains underexplored. In this study, we evaluated two PROTAC drugs, CC-90009 and GU3341, both of which target the degradation of the protein GSPT1. The treatment effectively inhibited tumor growth, suppressed cancer cell proliferation, and induced cell death. Notably, it showed pronounced efficacy in two subtypes of pediatric AML driven by specific gene fusions: RUNX1::RUNX1T1 and FUS::ERG. Additionally, the degradation of GSPT1 led to a reduction in the levels of these leukemia-causing gene fusions, highlighting the critical role of GSPT1 in the disease. These findings suggest a promising new approach for treating pediatric AML by targeting proteins essential for leukemic cell survival.

## 1. Introduction

Acute myeloid leukemia (AML) is a heterogeneous hematological malignancy characterized by uncontrolled clonal proliferation of myeloid progenitor cells in the bone marrow [1]. Despite intensive treatment and advanced supportive care, overall survival among children remains about 70%, with relapse rates ranging between 25 and 35% [2,3]. Moreover, survivors often suffer from severe side effects of the treatment with late sequelae [2,3]. This situation urgently calls for the development of novel therapies to improve the cure rate and quality of survivorship of pediatric AML.

Over the last decade, significant progress has been made in the molecular profiling of AML, providing a deeper understanding of its pathobiology and identification of therapeutic vulnerabilities [4]. However, targeting specific drivers of leukemogenesis is still one of the biggest challenges in both adult and pediatric AML. One promising therapeutic approach involves the selective degradation of target proteins, including fusion proteins, using proteolysis-targeting chimera (PROTAC) technology. PROTACs are bifunctional molecules consisting of a ligand that binds to an E3 ligase, connected by a linker to another ligand that binds to the protein of interest. By bringing the E3 ligase in the vicinity of the protein of interest, ubiquitination by the E3 ligase and subsequent proteasomal target degradation will be triggered [5,6]. Thanks to their unique mechanism of action, PROTACs can be developed to degrade proteins that do not contain targetable enzymatic protein binding functions with improved potency and reduced toxicity compared to previous small molecule inhibitors [7,8]. Several PROTACs have shown promising preclinical results in the treatment of hematological malignancies, including adult AML [9]. Nevertheless, this field is still poorly explored in pediatric AML, which often differs from adult AML in the kind of key driver mutation. Therefore, many of the target therapies developed for adult AML might have comparatively limited utility in children.

Among the targets with available PROTACs, cyclin-dependent kinase 6 (CDK6) holds significance in promoting the proliferation or viability of AML cells. CDK6 regulates the G1-S phase transition in the cell cycle and drives cell proliferation. CDK6 is overexpressed in adult AML and has been identified as a prognostic biomarker [10,11]. Furthermore, it has been implicated as a key regulator of leukemic stem cell (LSC) functions [12]. Several studies showed the high sensitivity of leukemic cells to palbociclib, a potent and specific inhibitor of CDK4/CDK6, already approved by the U.S. Food and Drug Administration for the treatment of HR-positive and HER2-negative breast cancer [13,14,15,16].

A second protein under consideration as an AML target is the G1 to S phase transition 1 protein (GSPT1). In contrast to CDK6, limited information is available regarding a direct link between GSPT1 and AML. Nonetheless, GSPT1 plays a central role in mRNA translation, and depletion of GSPT1 activates an integrated stress response that leads to cell death in leukemic cells while sparing normal hematopoietic stem cells and reduces leukemia engraftment and LSCs in primary adult AML patients [17,18]. PROTACs targeting GSPT1, such as CC-90009, have been tested in clinical trials as monotherapy (NCT02848001) or in combination with azacitidine, venetoclax, or gilteritinib (NCT04336982) for relapsed and refractory (R/R) adult AML.

To date, no published studies have explored the effectiveness of targeting CDK6 and GSPT1 in pediatric AML. In this study, we investigated the efficacies and molecular effects of CDK6 and GSPT1 degraders in two pediatric AML subtypes expressing either RUNX1::RUNX1T1, the most common fusion gene in pediatric AML, or FUS::ERG, which is associated with a very poor prognosis [14,19]. While PROTACs targeting only CDK6 degradation did not provide added benefit compared to the CDK4/CDK6 inhibitor palbociclib, PROTAC-mediated GSPT1 degradation yielded strong antiproliferative effects in both AML subtypes examined. Interestingly, this effect was accompanied by the depletion of the corresponding fusion genes both at RNA and protein levels, raising the possibility of GSPT1 controlling leukemic fusion gene expression. These data support GSPT1 as a promising target in pediatric AML, paving the way for new therapeutic interventions.

## 2. Materials and Methods

### 2.1. Cell Lines Culture

Kasumi-1 (RRID: CVCL_0589, ACC 220) and SKNO-1 (RRID: CVCL_2196, ACC 690) cells were maintained in Roswell Park Memorial Institute Medium (RPMI) 1640 supplemented with Glutamax (Gibco, Waltham, MA, USA) and 10% FBS (Sigma, St. Louis, MO, USA or 20% FBS and 8 ng/mL GM-CSF, respectively. TSU-1621-MT (RRID: CVCL_A455, female) and YNH-1 (RRID: CVCL_1927, male) cells were maintained in RPMI 1640 medium supplemented with 10% FBS and with 10 ng/mL GM-CSF. All cells were cultured at 37 °C in a humidified 5% CO_2_ incubator. Cells were regularly authenticated and tested for *Mycoplasma* negativity.

### 2.2. PDX and Primary Culture

Patient-derived xenografts (PDXs) and primary cells were co-cultured on mesenchymal stem cells (MSCs), as described previously [20]. MSCs were obtained from healthy human bone marrow and cultured in Dulbecco’s Modified Eagle Medium low glucose (Gibco) supplemented with 1% L-glutamine, 8 ng/mL recombinant human fibroblast growth factor-basic (Peptrotech, Cranbury, NJ, USA), 20% FBS, and 100 U/mL penicillin–streptomycin (P/S). PDX and primary cells were maintained in serum-free medium (Serum-Free Expansion Medium II, StemCell, Vancouver, BC, Canada) supplemented with 150 ng/mL stem cell factor, 100 ng/mL thrombopoietin, 10 ng/mL Fms-related tyrosine kinase 3 ligand, 1.35 µM UM729, 750 nM StemRegenin 1 (Biogeme, Lausanne, Switzerland), 10 ng/mL interleukin-3, 10 ng/mL granulocyte-macrophage colony-stimulating factor (all cytokines have been purchased from PeptroTech) and 100 U/mL P/S, at 37 °C in a humidified 5% CO_2_ incubator, as described previously [20]. All patients gave written consent for the use of their material for research purposes.

### 2.3. Drug Treatment

Cells were seeded for the drug treatment at a density of 5 × 10^5^ cells/mL, followed by drug or vehicle addition on the same day (day 0). Subsequent analyses were performed at different time points, as indicated in the respective figures. Drugs were renewed every 3 days. Appendix A shows the drugs used in this study. After drug treatment, cells were counted with 0.2% Trypan Blue in a hemocytometer (Hawksley, East Sussex, UK). The ED_50_ is defined as the concentration required for 50% inhibition of maximal cell proliferation.

### 2.4. Colony Formation Assay

Following treatment in suspension culture, cells were incubated for 8–11 days without compound until colonies grew to over 25 cells/colony before counting. To that end, cells were seeded in Methylcellulose (Sigma, St. Louis, MO, USA) media (0.56% (*w*/*v*) in complete medium containing 20% FBS) without compound at a density of 6000 cells/mL in a 24-well plate (3000 cells per well).

### 2.5. RNA Isolation and RT-qPCR

RNA was extracted using the Nucleospin RNA kit (Macherey-Nagel, Dueren, Germany) according to the manufacturer’s instructions. The RevertAid First Strand cDNA Synthesis Kit (Thermo Fisher, Waltham, MA, USA) was used according to the supplier’s protocol for reverse transcription. Real-time quantitative PCRs (RT-qPCR) were performed with the SsoAdvanced Universal SYBR Green Supermix (Biorad, 172-5275). The RT-PCRs were performed on a CFX384 qPCR machine (Biorad, Hercules, CA, USA). The qPCR primers are summarised in Appendix A.

### 2.6. Immunoblot Analysis

Proteins were isolated simultaneously with the RNA by precipitation of the Nucleospin RNA kit flow-through with two volumes of cold acetone and incubation for 15 min on ice. The protein pellets were then dissolved in Urea buffer (9 M urea, 4% CHAPS, 1% dithiothreitol) at a concentration of 25,000 cells/mL. Sample proteins were separated using SDS-PAGE electrophoresis and transferred onto a 0.45 μm methanol-activated PVDF membrane (Thermo Scientific, 88518). Membranes were incubated overnight with primary antibodies, listed in Appendix A. Then, membranes were washed and incubated with secondary antibodies for 1 h. As secondary antibodies were used: Polyclonal Goat Anti-Rabbit Immunoglobulins/HPR (Dako, Glostrup, Denmark) or Polyclonal Goat Anti-Mouse Immunoglobulins/HPR (Dako) at the dilution 1:5000. Protein/antibody complexes were detected by chemiluminescence, using Super Signal West Pico Plus Chemiluminescent Substrate (Thermo Scientific, 34580) at the FluorChemE imaging instrument (Proteinsimple, San Jose, CA, USA). The density of each protein band in Western blots was measured using ImageJ 1.53 software. Full Western blot images can be found in Appendix A.

### 2.7. Cell Cycle Analysis

Cells were washed twice in PBS and fixed by adding 2% PFA for 20 min on ice in the dark. After being washed twice with 1X PBS, cells were stained with 7.5 ug/mL Hoechst-33342 (Thermo Fisher, H3570) in permeabilization buffer (0.1% Sodium Citrate, 0.1% Triton X-100 in H_2_O) for 20 min at 37 °C, protected from light. The analysis was performed at CytoFLEX LX Flow Cytometry (Beckman Coulter, Brea, CA, USA).

### 2.8. Apoptosis Assay

Cells were washed in Annexin V Binding Buffer (10 mM HEPES, 140 mM NaCl, 2.5 mM CaCl_2,_ pH = 7.4) followed by incubation with Annexin V (BioLegend, San Diego, CA, USA) and SYTOX^TM^ Deep Red Nucleic Acid Stain (Invitrogen, Waltham, MA, USA) for 15 min at 37 °C. The number of early apoptotic cells, defined as Annexin V-positive and SYTOX-negative cells, and the number of late apoptotic/necroptotic cells, defined as Annexin V-positive and SYTOX-positive cells, was determined by CytoFLEX LX Flow Cytometry (Beckman Coulter).

### 2.9. Statistical Analysis

Data were analyzed with GraphPad Prism 8 (GraphPad Software) using a one-way ANOVA. The variance of biological replicates is represented as the standard deviation (SD). Differences with *p*-value ≤ 0.05 were recognized statistically. * *p* < 0.05; ** *p* < 0.01; *** *p* < 0.001; **** *p* < 0.0001.

## 3. Results

### 3.1. GU3341 PROTAC Induces Stronger Anti-AML Activity in RUNX1::RUNX1T1 Cell Lines Compared to Palbociclib

Previous genome-wide CRISPR screens suggested a reliance of several AML cell lines on CDK6 but not its close homolog CDK4 [21]. Further work demonstrated that the propagation of AML subtypes expressing RUNX1::RUNX1T1, *KMT2A*, and *NUP98* fusion genes or *FLT3-ITD* is dependent on CDK6 [14,15,16,22].

To further investigate the reliance of RUNX1::RUNX1T1 AML on CDK6, we examined the effect of three CDK6 PROTACs on the proliferation of AML cell lines. Specifically, we tested a von Hippel–Lindau (VHL)-based PROTAC named CST651 and two cereblon (CRBN)-based PROTACs, named GU3341 and BSJ-03-123 [23] (Figure 1A), and with the latter already shown to cause a robust anti-proliferative effect in CDK6-dependent AML cell lines [21]. All PROTACs contain the CDK4/6 inhibitor palbociclib as a CDK6 targeting ligand. We investigated the impact of these PROTACs on the proliferation and viability of two RUNX1::RUNX1T1 cell lines, Kasumi-1 and SKNO-1, both expressing transcripts coding for the E3 ligases *CRBN* and *VHL* (Figure 1B). To evaluate the impact of CDK6 degradation, we compared the effect of PROTACs with palbociclib. GU3341 showed potent antiproliferative activity in both Kasumi-1 and SKNO-1 cell lines (Figure 1C and Appendix A). Furthermore, GU3341 showed a stronger cytotoxic effect than palbociclib, inducing around a 90% kill rate at a dose of 1000 nM, compared to a 55% kill rate induced by palbociclib at a similar concentration in both cell lines (Figure 1D). GU3341 pretreatment yielded substantial inhibition in colony formation, causing a 50% reduction in colony number at the concentration of 100 nM, while palbociclib only caused around a 20% reduction of colony formation at the same concentration (Figure 1E,F). In contrast, the degradation of CDK6 by BSJ-03-123 or CST651 did not cause a similar antiproliferative effect as palbociclib or the PROTAC GU3341 (Figure 1B–F). These data suggest that CDK6-PROTACs with different chemical structures have distinct antiproliferative effects in RUNX1::RUNX1T1 AML cell lines.

### 3.2. CDK6-PROTACs Preferentially Reduce CDK6 Protein Levels

To test whether CDK6-PROTACs selectively suppress CDK6 protein levels and not those of its homolog CDK4, immunoblotting analysis was performed in Kasumi-1 cells treated with different concentrations of CDK6-PROTACs. BSJ-03-123 and CST651, at 100 and 1000 nM concentrations, reduced CDK6 protein levels by 30% and 70%, respectively, after 24 h of treatment. However, after 72 h of BSJ-03-123 or CST651 treatment, CDK6 protein was again highly expressed in Kasumi-1 cells, suggesting that this degradation is not maintained over time (Figure 2A,B and Appendix A). In contrast, treatment with 100 nM and 1000 nM of PROTAC GU3341 significantly decreased CDK6 levels by 70% and 85%, respectively, after 24 h of treatment and by 75% and 90%, respectively, after 72 h of treatment. Simultaneously, following treatment with GU3341 for 24 or 48 h, CDK4 underwent degradation at both 100 nM and 1000 nM. Nevertheless, CDK4 protein levels partially recovered after 72 h despite the continued presence of the PROTAC (Figure 2C and Appendix A). As a consequence of CDK6 knockdown, CST651 and GU3341 treatment also reduced the CDK6-mediated phosphorylation of S780 of the retinoblastoma (RB1) protein (Figure 2D,E and Appendix A), a tumor suppressor that blocks cell cycle progression when unphosphorylated. These combined data confirm GU3341 as the most potent of the three CDK6 PROTAC tested, possibly explaining the reduced inhibition of leukemic proliferation.

### 3.3. GU3341 Reduces GSPT1 and Ikaros Protein Levels in AML Cells

Because of its superior antiproliferative effect and more efficient CDK6 degradation compared to the other two PROTACs, we examined the influence of GU3341 on the cell cycle. Treatment of Kasumi-1 cells with GU3341 reduced the fraction of cells in the S phase two-fold (DMSO: 30% cells in S-phase; 1000 nM GU3341: 15% cells in S-phase) and increased the percentage of cells in the sub-G1 phase ten-fold (DMSO: 6% subG1 cells; 1000 nM GU3341: 61% subG1 cells) (Figure 3A,B and Appendix A). This increase correlates with a higher incidence of apoptotic cells following GU3341 treatment compared to palbociclib treatment (Figure 3C and Appendix A).

GU3341 is a CRBN-based PROTAC, derived from thalidomide. Thalidomide induces degradation of proteins such as the zinc finger transcription factors Ikaros (IKZF1) and Aiolos (IKZF3), which are regulators of hematopoietic lineage commitment, or such as the translation termination factor G1 to S phase transition protein 1 (GSPT1) or casein kinase 1α (CK1α) [24]. We, therefore, examined whether GU3341 triggers the degradation of proteins other than CDK6. In agreement with other thalidomide-based PROTACs, IKZF1 and GSPT1 protein levels were only diminished in Kasumi-1 cells after GU3341 treatment but not by incubation with the other two PROTACs (Figure 3D and Appendix A). These findings indicate that the potent antiproliferative effect of GU3341 on AML cell lines may arise not only from CDK6 degradation but also from an off-target degradation of GSPT1 and IKZF1 proteins.

### 3.4. Degradation of GSPT1 Impairs RUNX1::RUNX1T1 Expression

Previous findings showed that depletion of GSPT1 by the molecular glue degrader CC-90009, a cereblon E3 ligase modulating drug that coopts CRL4^CRBN^ to selectively target GSPT1 ubiquitination and proteasomal degradation (Figure 4A), rapidly induces AML apoptosis and reduces leukemia engraftment [17]. These and our data suggest that the degradation of GSPT1 contributes to the antileukemic effect of GU3341.

We, therefore, examined the antiproliferative effect of CC-90009 in RUNX1::RUNX1T1 Kasumi-1 cells. CC-90009 showed a potent cytotoxic activity with an ED_50_ of 34.1 ± 7.8 nM after 24 h, which decreases over time of exposure to 19.4 ± 8.9 nM and 8.1 ± 2.1 nM after 48 and 72 h of treatment, respectively (Figure 4B and Appendix A). Through immunoblotting, we verified the targeted degradation of GSPT1 by CC-90009. Unlike GU3341, after 24 h of treatment, CC-90009 specifically induced full degradation of GSPT1 protein and not IKZF1, another CRBN-based PROTAC target (Figure 4C and Appendix A).

CC-90009 treatment reduced RUNX1::RUNX1T1 expression both at protein and mRNA levels. Treatment with 100 nM and 1000 nM CC-90009 diminished RUNX1::RUNX1T1 protein 2.5-fold after 24 h and almost completely after 48 h of treatment. Interestingly, changes in *RUNX1::RUNX1T1* transcript levels seem to trail protein levels with only 30% and 50% decreases at 24 and 48 h of treatment, respectively (Figure 4D,E and Appendix A). Additionally, we observed a significant decrease in the expression of the wild-type *RUNX1* and *ERG* genes, both of which are essential transcription factors cooperating with RUNX1::RUNX1T1 [25,26,27]. (Figure 4F,G). RUNX1 protein levels were diminished by 60% and fully eliminated after 48 h of treatment at concentrations of 100 nM and 1000 nM, respectively, again with less reduced transcript levels (Figure 4D). In contrast, ERG transcript and protein levels showed more comparable changes with complete loss of protein upon 48 h treatment with 10 nM of CC-90009 (Figure 4H and Appendix A). These data suggest that GSPT1 degradation interferes with leukemic gene expression networks by the combined elimination of RUNX1, ERG, and RUNX1::RUNX1T1.

### 3.5. CC-90009 Treatment Impairs RUNX1::RUNX1T1 Expression in Primary AML Cells

To further assess the antileukemic activity of CC-90009 we treated RUNX1::RUNX1T1-positive PDX and primary AML cells with CC-90009 over more extended periods. To that end, we cultured PDX on human bone marrow-derived primary mesenchymal stromal cells to support AML proliferation and viability and maintain immature AML cell population [20]. CC-90009 inhibited strongly the proliferation of RUNX1::RUNX1T1 PDX and primary cell already at low doses of 10 nM and 3 nM, respectively (Figure 5A,B, Appendix A). This was associated with a block in cell cycle progression and an increase in the fraction of apoptotic cells in the subG1 phase (Figure 5C and Appendix A). Dose–response experiments determined ED_50_ values of 50 nM and 21 nM at 24 and 48 h, respectively (Figure 5D and Appendix A). Furthermore, CC-90009 induced complete elimination of RUNX1::RUNX1T1 protein after 48 h of treatment at ≥100 nM (Figure 5E and Appendix A). Kinetics of changes in *RUNX1::RUNX1T1*, *RUNX1* and *ERG* transcript levels paralleled those of the corresponding protein levels (Figure 5F–H). These data confirmed the anti-proliferative effect of CC-90009 in RUNX1::RUNX1T1 PDX and primary cells and the induction of fusion protein depletion in AML PDX cells.

### 3.6. CC-90009 and GU3341 PROTACs Induce Anti-AML Activity in FUS::ERG Cell Lines

The significant degradation of ERG induced by CC-90009 prompted us to examine the impact of this PROTAC on FUS::ERG-positive t(16;21)(p11;q22) AML, which is associated with a particularly poor clinical outcome [19]. We tested the effect of GSPT1 degradation in two FUS::ERG AML cell lines, TSU-1621-MT and YNH-1, both expressing significant *CRBN* levels (Figure 6A). CC-90009 PROTAC showed a stronger antiproliferative activity in TSU-1621-MT when compared with that of Kasumi-1 cells (ED_50_ of 30.2 ± 13 after 24 h of treatment, ED_50_ of 2.0 ± 0.8 after 48 h of treatment) (Figure 6B,C and Appendix A). After 72 h of treatment, CC-90009 showed a potent cytotoxic effect already at the low dose of 10 nM (Figure 6C). Interestingly, we observed a significant decrease in *FUS::ERG* fusion transcript levels after 24 h treatment with CC-90009 (Figure 6D). Through immunoblotting we confirmed that GU3341 PROTAC, a CDK6 PROTAC, induces GSPT1 degradation in both t(16,21) cell lines tested in this study (Figure 6E and Appendix A). Furthermore, GU3341 PROTAC showed a strong cytotoxic effect in TSU-1621-MT and YNH-1 cell lines (Figure 6F–H). GU3341 induced more than 90% killing already at 100 nM and almost 100% killing at 1000 nM after 72 h treatment, in both cell lines (Figure 6G,H and Appendix A). Similar to CC-90009, both FUS::ERG cell lines were 5–10-fold more sensitive towards GU3341 when compared to RUNX1::RUNX1T1 cell lines (ED_50_ Kasumi-1 = 164 ± 2 nM, ED_50_ TSU-1621-MT= 16 ± 2 nM, ED_50_ YNH-1= 29 ± 2 nM). These data suggest that t(16;21)(p11;q22) AML is particularly sensitive to GSPT1 degraders.

## 4. Discussion

The substantial side effects associated with chemotherapy highlight the necessity to find new, at least as effective, and more targeted therapeutic strategies for AML treatment. This study investigated the potential of CDK6-targeting PROTACs as a novel and potentially powerful strategy to improve the treatment of pediatric AML. Our data show that off-target degradation of GSPT1, together with CDK6 degradation, induces a strong antileukemic effect and that GSPT1 loss is linked to impaired expression of the *RUNX1::RUNX1T1* and *FUS::ERG* fusion genes.

From the investigation of CDK6-PROTACs, all three compounds tested in this study exhibited anti-proliferative effects in RUNX1::RUNX1T1 AML cell lines. However, only GU3341 PROTAC was shown to have a more potent effect than the dual CDK4/CDK6 inhibitor palbociclib, which formed the warhead in all three PROTACs examined. Interestingly, the strong effect of GU3341 in inhibiting proliferation and inducing apoptosis in AML cells is at least partially due to the degradation of the protein GSPT1.

Our findings that GSPT1 degradation affects the levels of RUNX1 and ERG, in addition to the RUNX1::RUNX1T1 and FUS::ERG fusion genes, suggest a reorganization of leukemic transcriptional networks. Both RUNX1 and ERG are members of a transcription factor heptad controlling normal hematopoiesis [28]. Moreover, RUNX1::RUNX1T1 AML cells [25] are addicted to ERG and RUNX1 with a disbalanced expression in favor of RUNX1::RUNX1T1 causing leukemia cell death. Instead, two studies pointed at deregulation of the mTOR pathway and induction of an integrated stress response as apoptotic triggers [17,18]. To what extent the loss of fusion gene expression contributes to the observed antileukemic effects remains subject to further research.

Previous studies have reported a strong anti-leukemic effect of GSPT1 [17,18] degradation for different AML subtypes. However, clinical trials involving CC-90009 have been recently terminated in R/R adult AML patients, due to lack of efficacy as a single agent in the short-term acute phase. The lack of efficacy as a single agent might be linked to a molecularly heterogeneous population of AML patients that had undergone multiple rounds of chemotherapy. Since final patient outcomes have not been published yet, it remains unclear whether the patient cohort included patients of the two subtypes investigated in this study, thus leaving the possibility that certain subsets of pediatric and adolescent AML might actually benefit from CC-90009 treatment.

In this study, we found surprising anti-proliferation efficacy in de novo pediatric AML subtype characterized by RUNX1::RUNX1T1 fusion gene. Interestingly, treatment with CC-90009 PROTAC was associated with a potent degradation of RUNX1::RUNX1T1 both at transcript and protein levels, suggesting that GSPT1 regulates the expression of this leukemic fusion gene. Additionally, we found that upon GSPT1 degrader treatment, wild-type RUNX1 and ERG levels are diminished both at protein and transcript levels. In line with the reduced ERG expression, both CC-90009 and GU3341 PROTACs demonstrated a potent antiproliferative effect in FUS::ERG AML cells, associated with reduced *FUS::ERG* fusion transcript levels. The strong degradation by GSPT1 degraders may also explain the increased sensitivity of FUS::ERG AML cells compared to RUNX1::RUNX1T1 cells. Together, these data suggest two possibly non-exclusive hypotheses. Given GSPT1’s established role in translational termination, GSPT1 might either directly control the translation of these transcriptional regulators or enhance the translation of factors driving their expression. Further research is needed to experimentally test these two models.

Taken together, our study provides compelling evidence that GSPT1 degradation leads to a strong anti-proliferative effect in RUNX1::RUNX1T1 AML subgroup and to the corresponding degradation of the RUNX1::RUNX1T1 fusion protein, as well as a robust antiproliferative activity in the high-risk FUS::ERG AML subgroup. Overall, these findings suggest that targeting GSPT1 could be a promising therapeutic strategy for certain pediatric AML subtypes and point to a possible role in the regulation of the stability and translation of critical fusion proteins such as RUNX1::RUNX1T1 or FUS::ERG.

## 5. Conclusions

Our study highlights the therapeutic potential of targeting GSPT1 protein, particularly in pediatric AML subtypes characterized by RUNX1::RUNX1T1 and FUS::ERG fusion genes. The findings demonstrated that GSPT1 degradation disrupts leukemic transcriptional networks by reducing the expression of critical fusion protein and associated transcriptional factors, resulting in a strong antiproliferative effect. These results suggest that GSPT1-targeting strategies could provide a novel and more targeted therapeutic option for RUNX1::RUNX1T1 and FUS::ERG AML subgroups, warranting further clinical exploration.

## Figures and Tables

**Figure 1 cancers-17-00211-f001:**
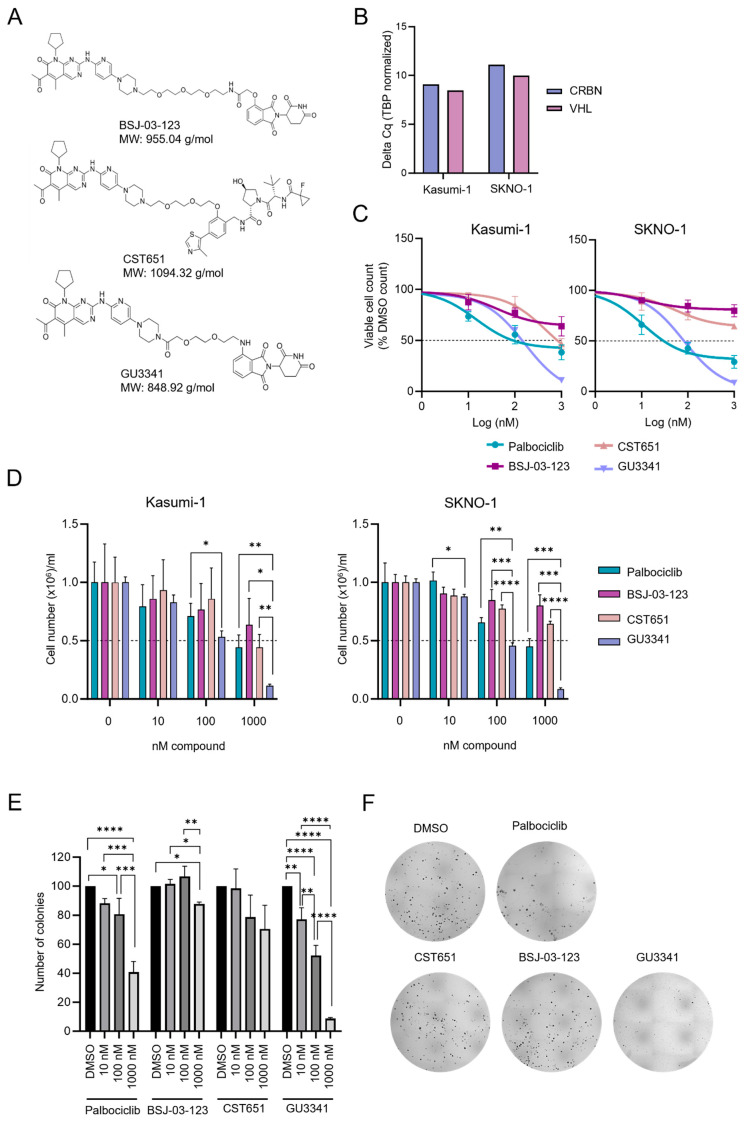
GU3341 PROTAC induces stronger anti-AML activity in RUNX1::RUNX1T1 cell lines compared to palbociclib. (**A**) The chemical structure of CDK6-PROTACs: BSJ-03-123, CST651, and GU3341. (**B**) Quantitative PCR analysis of the expression levels of CRBN and VHL in Kasumi-1 and SKNO-1 cell lines (*n* = 1 biologically independent sample). (**C**) Proliferation curve of Kasumi-1 cell line treated with palbociclib, BSJ-03-123, CST651, and GU3341 for 72 h (mean ± SD, *n* = 3 biologically independent samples). (**D**) Viability of Kasumi-1 and SKNO-1 cell lines treated with palbociclib, BSJ-03-123, CST651, GU3341, and DMSO for 72 h (mean ± SD, *n* = 3 biologically independent samples). Significant *p*-values were plotted to compare differences between drug-treatment groups at the same dose. (**E**) Number of colonies of Kasumi-1 cell line treated with palbociclib, BSJ-03-123, CST651, GU3341, and DMSO for 72 h (mean ± SD, *n* = 3 biologically independent samples). (**F**) Images of colony formation assay of Kasumi-1 cell line treated with 1000 nM palbociclib, BSJ-03-123, CST651, GU3341, and DMSO for 72 h. * (*p* < 0.05), ** (*p* < 0.01), *** (*p* < 0.001), **** (*p* < 0.0001) indicate differences between treatment groups.

**Figure 2 cancers-17-00211-f002:**
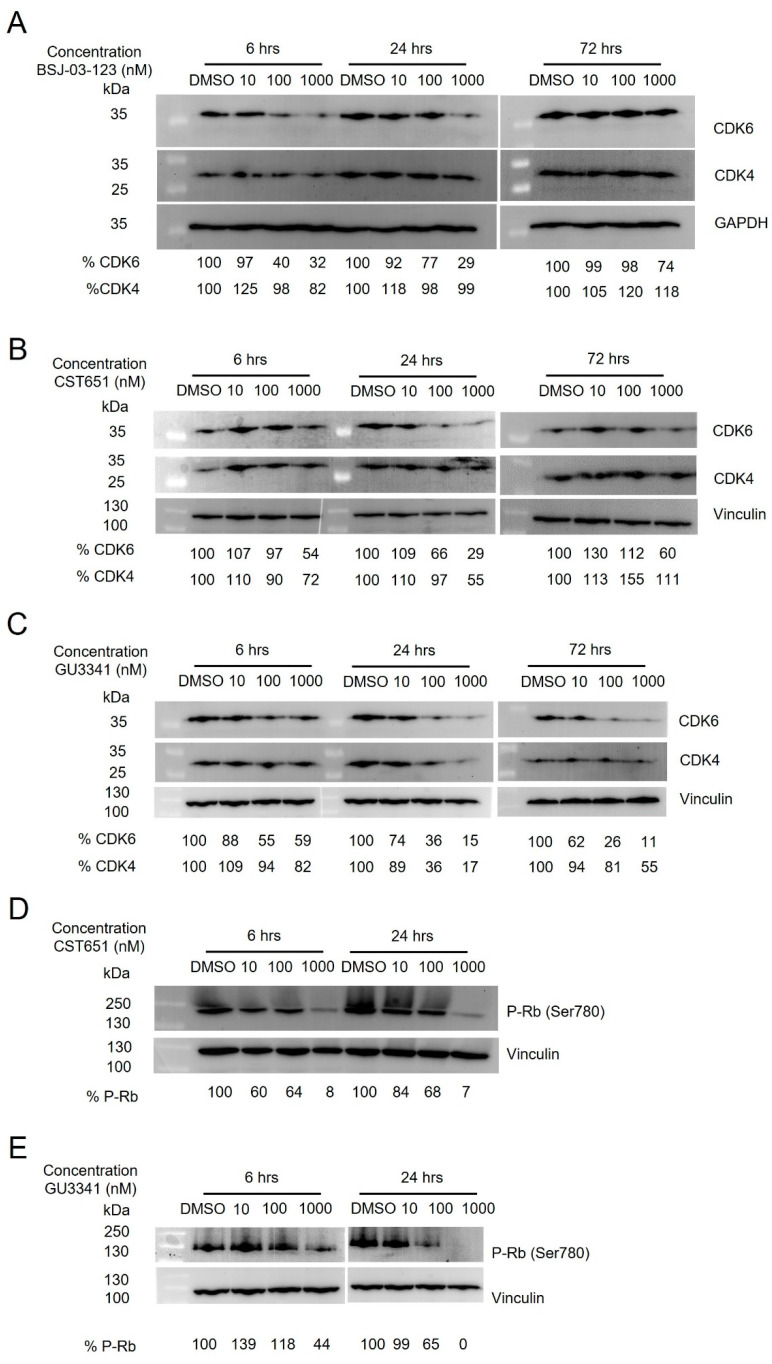
CDK6-PROTACs preferentially reduce CDK6 protein levels. (**A**) Western blotting of CDK6, CDK4, and GAPDH in Kasumi-1 cell line treated with BSJ-03-123 or DMSO for 24, 48, and 72 h. (**B**) Western blotting of CDK6, CDK4, and vinculin in Kasumi-1 cells treated with CST651 or DMSO for 24, 48, and 72 h. (**C**) Western blotting of CDK6, CDK4, and vinculin in Kasumi-1 cells treated with GU3341 or DMSO for 24, 48, and 72 h. (**D**) Western blotting of P-Rb (Ser780) and vinculin in Kasumi-1 cells treated with CST651 or DMSO for 6 and 24 h. (**E**) Western blotting of P-Rb (Ser780) and vinculin in Kasumi-1 cells treated with GU3341 or DMSO for 6 and 24 h.

**Figure 3 cancers-17-00211-f003:**
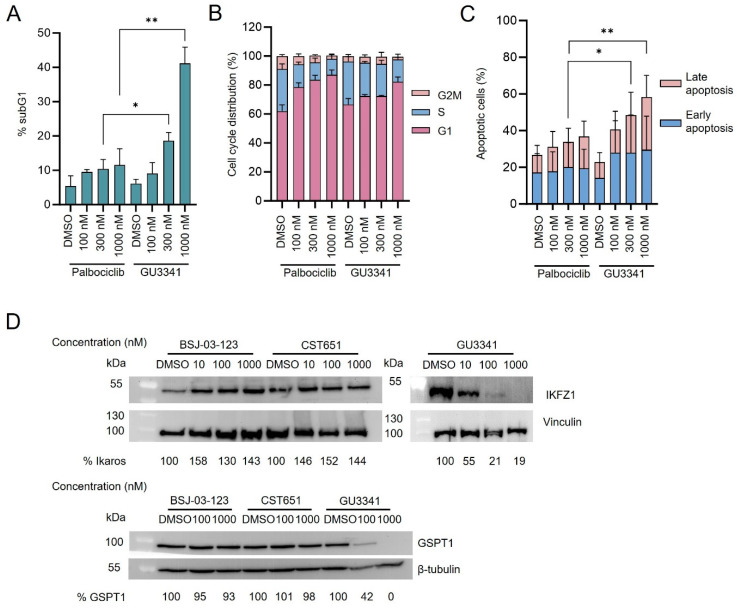
GU3341 reduces GSPT1 and Ikaros protein levels in RUNX1::RUNX1T1 AML cells. (**A**) Percentage of cells in subG1 phase of the cell cycle in Kasumi-1 cells after treatment with palbociclib or GU3341 or DMSO for 72 h (mean ± SD, *n* = 3 biologically independent samples). Significant *p*-values were plotted to compare differences between drug-treatment groups at the same dose. (**B**) Percentage of cells in G1/S/G2M phases of cell cycle in Kasumi-1 cells after treatment with palbociclib or GU3341 or DMSO for 72 h (mean ± SD, *n* = 3 biologically independent samples). (**C**) Apoptosis assay Sytox Red in Kasumi-1 cells after treatment with palbociclib or GU3341 or DMSO for 72 h (mean ± SD, *n* = 3 biologically independent samples). Significant *p*-values were plotted to compare differences between drug-treatment groups at the same dose. (**D**) Western blotting of GSPT1, Ikaros, β-tubulin, vinculin in Kasumi-1 cells treated with BSJ-03-123, CST651, GU3341, or DMSO for 24 h. * (*p* < 0.05) and ** (*p* < 0.01) indicate differences between treatment groups.

**Figure 4 cancers-17-00211-f004:**
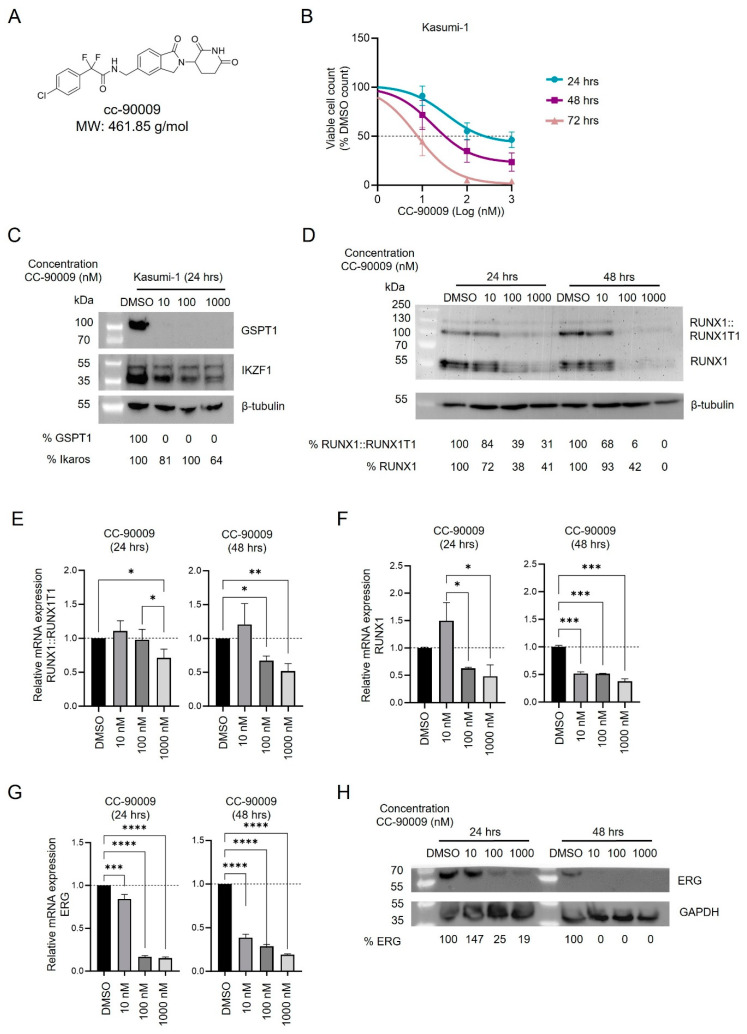
CC-90009 induces anti-AML activity in a RUNX1::RUNX1T1 cell line. (**A**) The chemical structure of CC-90009. (**B**) Proliferation curve of Kasumi-1 cell line treated with CC-90009 or DMSO for 24, 48, and 72 h (mean ± SD, *n* = 3 biologically independent samples). (**C**) Western blotting of GSPT1, Ikaros, and β-tubulin in Kasumi-1 cells treated with CC-90009 or DMSO for 72 h. (**D**) Western blotting of RUNX1::RUNX1T1, RUNX1 and β-tubulin in Kasumi-1 cells treated with CC-90009 or DMSO for 24 and 48 h. (**E**) Quantitative PCR analysis of the expression levels of RUNX1::RUNX1T1 transcripts in Kasumi-1 cells treated with CC-90009 or DMSO for 24 and 48 h (mean ± SD, *n* = 3 biologically independent samples). (**F**) Quantitative PCR analysis of the expression levels of RUNX1 transcripts in Kasumi-1 cells treated with CC-90009 or DMSO for 24 and 48 h (mean ± SD, *n* = 3 biologically independent samples). (**G**) Quantitative PCR analysis of the expression levels of ERG transcripts in Kasumi-1 cells treated with CC-90009 or DMSO for 24 and 48 h (mean ± SD, *n* = 3 biologically independent samples). (**H**) Western blotting of ERG and GAPDH in Kasumi-1 cells treated with CC-90009 or DMSO for 24 and 48 h. * (*p* < 0.05), ** (*p* < 0.01), *** (*p* < 0.001), **** (*p* < 0.0001) indicate differences between controls and treatment groups.

**Figure 5 cancers-17-00211-f005:**
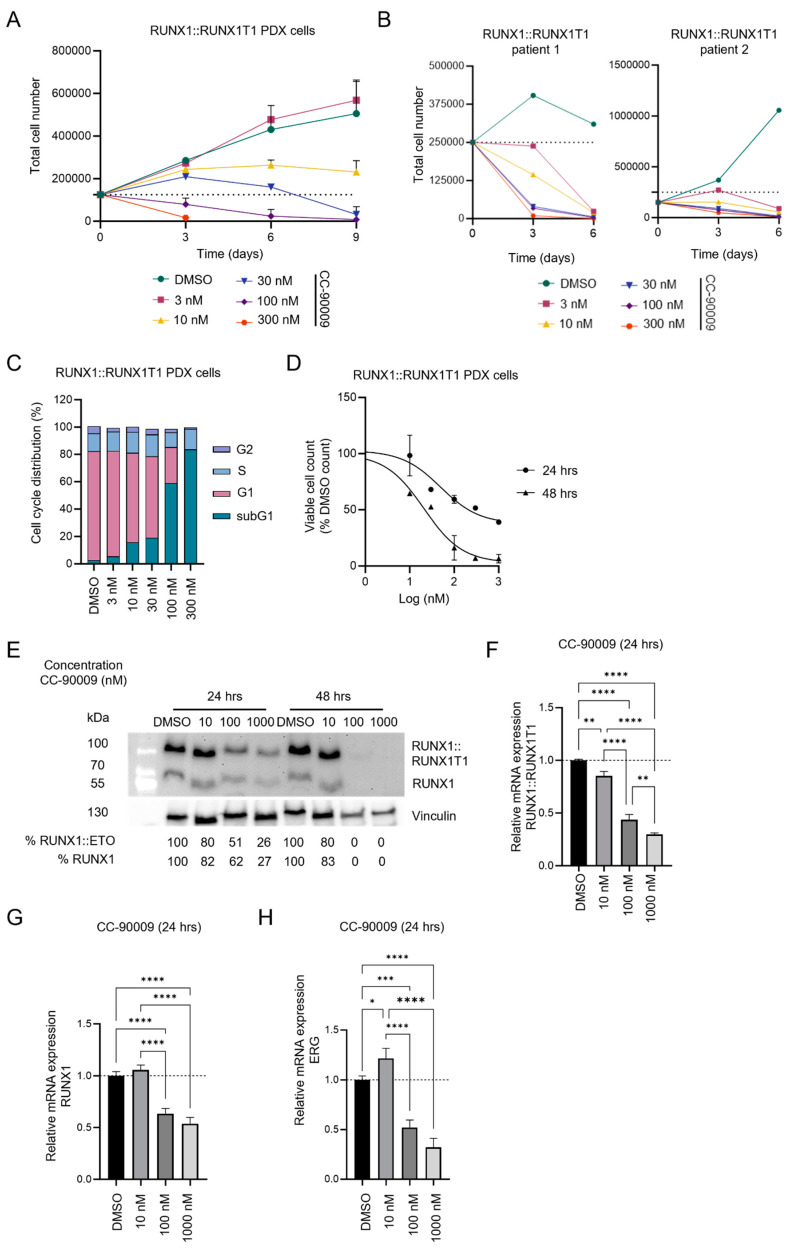
CC-90009 treatment reduces RUNX1::RUNX1T1 expression in PDX and primary AML cells. (**A**) Proliferation curve of RUNX1::RUNX1T1 PDX cells treated with CC-90009 or DMSO. Cells were counted and treatment was refreshed every 3 days (mean ± SD, *n* = 3 biologically independent samples). (**B**) Proliferation curve of primary RUNX1::RUNX1T1 AML cells treated with CC-90009 or DMSO. Cells were counted and treatment was refreshed every 3 days (*n* = 2 biologically independent samples). (**C**) Cell cycle analysis after treatment with CC-90009 or DMSO for 3 days (*n* = 1 biologically independent samples). (**D**) Proliferation curve of RUNX1::RUNX1T1 PDX cells treated with CC-90009 or DMSO for 24 and 48 h (mean ± SD, *n* = 3 biologically independent samples). (**E**) Western blotting of RUNX1::RUNX1T1, RUNX1, and vinculin in RUNX1::RUNX1T1 PDX cells treated with CC-900009 or DMSO for 24 and 48 h. (**F**) Quantitative PCR analysis of the expression level of RUNX1::RUNX1T1 transcripts in RUNX1::RUNX1T1 PDX cells treated with CC-90009 or DMSO for 24 h (mean ± SD, *n* = 3 biologically independent samples). (**G**) Quantitative PCR analysis of the expression of RUNX1 transcript in RUNX1::RUNX1T1 PDX cells treated with CC-900009 or DMSO for 24 h (mean ± SD, *n* = 3 biologically independent samples). (**H**) Quantitative PCR analysis of the expression of ERG transcript in RUNX1::RUNX1T1 PDX cells treated with CC-900009 or DMSO for 24 h (mean ± SD, *n* = 3 biologically independent samples). * (*p* < 0.05), ** (*p* < 0.01), *** (*p* < 0.001), **** (*p* < 0.0001) indicate differences between controls and treatment groups.

**Figure 6 cancers-17-00211-f006:**
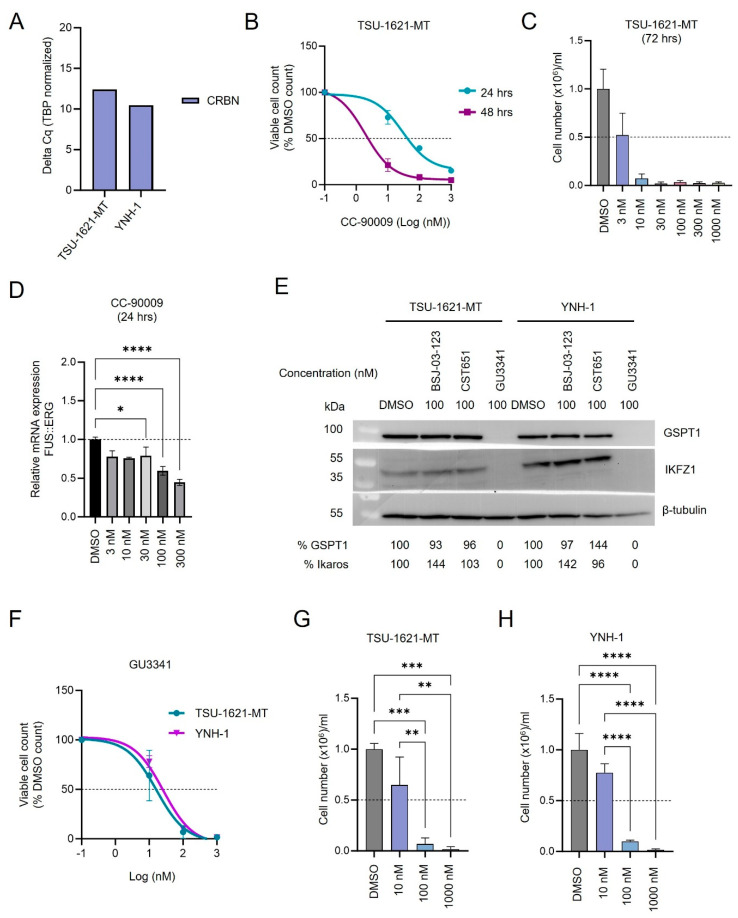
CC-90009 and GU3341 PROTACs induce anti-AML activity in FUS::ERG cell lines. (**A**) Quantitative PCR analysis of the expression levels of CRBN in TSU-1621-MT and YNH-1 cell lines (*n* = 1 independent experiment). (**B**) Proliferation curve of TSU-1621-MT cell line treated with CC-90009 and DMSO for 24 and 48 h (mean ± SD, *n* = 3 biologically independent samples). (**C**) Viability of TSU-1621-MT cells treated with CC-90009 and DMSO for 72 h (mean ± SD, *n* = 3 biologically independent samples). (**D**) Quantitative PCR analysis of the expression level of FUS::ERG transcripts in TSU-1621-MT cells treated with CC-90009 or DMSO for 24 h (mean ± SD, *n* = 3 biologically independent samples). (**E**) Western blotting of GSPT1, Ikaros, β-tubulin, vinculin in TSU-1621-MT and YNH-1 cells treated with BSJ-03-123, CST651, GU3341, or DMSO for 24 h. (**F**) Proliferation curve of TSU-1621-MT and YNH-1 cell lines treated with GU3341 and DMSO for 72 h (mean ± SD, *n* = 3 biologically independent samples). (**G**) Viability of TSU-1621-MT cells treated with GU3341 and DMSO for 72 h (mean ± SD, *n* = 3 biologically independent samples). (**H**) Viability of YNH-1 cells treated with GU3341 and DMSO for 72 h (mean ± SD, *n* = 3 biologically independent samples). * (*p* < 0.05), ** (*p* < 0.01), *** (*p* < 0.001), **** (*p* < 0.0001) indicate differences between controls and treatment groups.

## Data Availability

The data presented in this study are available in this article (and Appendix A).

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
