# Peer review of "PROTAC-Mediated GSPT1 Degradation Impairs the Expression of Fusion Genes in Acute Myeloid Leukemia"

_cancers, 2025, doi:10.3390/cancers17020211_

Round 1

Reviewer 1 Report

Comments and Suggestions for Authors

The manuscript demonstrates the application of PROTACs tageting GSPT1 in pediatric AML, signifying a novel strategy for cancer therapy. The effects of fusion genes, including RUNX1::RUNX1T1 and FUS::ERG were comprehensively investigated in both in vitro and ex vivo contexts. In summary, this paper is well-organized, featuring a thorough approach and clear results, and a minor revision will be suggested.

1.    English proficiency can be improved, particularly in the summary and abstract portions, to facilitate audience comprehension of this paper. 

2.    In Figures 1B and 6A Why is there just one data point for PCR analysis? As a critical aspect of this paper, it is recommended to utilize at least n=3 to enhance statistical dependability.

3.    A schematic illustration is preferred to emphasize the principal concept of this article.

4.    Figures 5a and 5b can utilize distinct colors to highlight and differentiate between individual groups.

Author Response

The manuscript demonstrates the application of PROTACs targeting GSPT1 in pediatric AML, signifying a novel strategy for cancer therapy. The effects of fusion genes, including RUNX1::RUNX1T1 and FUS::ERG were comprehensively investigated in both in vitro and ex vivo contexts. In summary, this paper is well-organized, featuring a thorough approach and clear results, and a minor revision will be suggested.

Comment 1: English proficiency can be improved, particularly in the summary and abstract portions, to facilitate audience comprehension of this paper.

Response 1: Thank you for pointing this out. As suggested, we have revised the summary and abstract to improve English proficiency and enhance clarity. In the rewritten version, we underlined the part that are replaced.

The simple summary (lines 12-22) has been replaced with the following text:

“Proteolysis targeting chimeras (PROTACs) are an innovative and promising technology designed to degrade harmful proteins associated with cancer. While these drugs have demonstrated significant potential in treating various cancer types, their application in pediatric acute myeloid leukemia (AML) remains underexplored. In this study, we evaluated two PROTAC drugs, CC-90009 and GU3341, both of which target the degradation of the protein GSPT1. The treatment effectively inhibited tumor growth, suppressed cancer cell proliferation, and induced cell death. Notably, it showed pronounced efficacy in two subtypes of pediatric AML driven by specific gene fusions: RUNX1::RUNX1T1 and FUS::ERG. Additionally, the degradation of GSPT1 led to a reduction in the levels of these leukemia-causing gene fusions, highlighting the critical role of GSPT1 in the disease. These findings suggest a promising new approach for treating pediatric AML by targeting proteins essential for leukemic cell survival.”

The abstract (lines 23-38) has been replaced with the following text:

“Proteolysis targeting chimeras (PROTACs) are heterobifunctional small molecules that utilize the ubiquitin-proteasome system to selectively degrade target proteins. This innovative technology has shown remarkable efficacy and specificity in degrading oncogenic proteins and has progressed through various stages of preclinical and clinical development for hematologic malignancies, including adult acute myeloid leukemia (AML). However, the application of PROTACs in pediatric AML remains largely unexplored. In this study, we show the potent effect of GSPT1 degradation against AML cells induced by either a GSPT1-selective cereblon modulator CC-90009 or by an off-target effect caused by a CDK6-PROTAC named GU3341. Both in vitro and ex vivo experiments revealed that GSPT1 degradation significantly inhibited tumor growth, induced cell cycle arrest, and triggered apoptosis in two pediatric AML subtypes characterized by RUNX1::RUNX1T1 and FUS::ERG fusion genes. Furthermore, the degradation of GSPT1 impaired the expression of RUNX1::RUNX1T1 and its cooperating transcription factors RUNX1 and ERG. Similarly, GSPT1 degradation also reduced FUS::ERG fusion transcript levels in AML cells harboring the translocation t(16;24)(p11:q22). These findings suggest a new role of GSPT1 in regulating leukemic transcriptional networks and open a new therapeutic strategy to target leukemic fusion genes in pediatric AML patients.”

Additionally, we replaced the following sentence throughout the entire text to enhance English proficiency:

  • Line 52: Changed "leading to a more profound comprehension" to "provided a deeper understanding".
  • Line 89-90: Changed "PROTACs leading only to CDK6 degradation did not show added value compared to" to "PROTACs targeting only CDK6 degradation did not provide added benefit compared to".
  • Line 175: Changed “addiction” with “reliance”.
  • Line 389- 390: Changed “all three different compounds tested in this study had an anti-proliferative effect in RUNX1::RUNX1T1 AML cell lines” to “all three compounds tested in this study exhibited anti-proliferative effects in RUNX1::RUNX1T1 AML cell lines.”
  • Line 395-396: Replaced “observation” with findings” and “points to” with “suggest”.
  • Line 422-423: Changed “These combined data suggest two possibly non-exclusive hypotheses" to “Together, these data suggest two potentially non-exclusive hypotheses."
  • Line 426: Changed “Further research will be required to experimentally test these two models.” to “Further research is needed to experimentally test these models."

Comment 2: In Figures 1B and 6A Why is there just one data point for PCR analysis? As a critical aspect of this paper, it is recommended to utilize at least n=3 to enhance statistical dependability.

Responses 2: We thank you the reviewer for pointing this out. In Figure 1B and 6A, we performed quantitative PCR analysis to detect the expression levels of CRBN and VHL in all the cell lines used in this study (Kasumi-1, SKNO-1, TSU-1621-MT and YHN-1). We believe that a single biological replicate is sufficient in this context, as all experiments were performed using the same batch of cell lines, thereby minimizing biological variability. These analyses were conducted in technical triplicates to ensure accuracy and reproducibility.

Comment 3: A schematic illustration is preferred to emphasize the principal concept of this article.

Response 3: We agree with this comment. We included a graphical abstract to emphasize the principal concept of this article.

Comment 4: Figures 5a and 5b can utilize distinct colors to highlight and differentiate between individual groups.

Response 4: Thank you for pointing this out. We agree with this comment and included colours in Figure 5A and 5B. 

Reviewer 2 Report

Comments and Suggestions for Authors

The manuscript is an interesting report showing that targeting the GSPT1 protein with two different PROTAC compounds decreases leukemic cellular survival and proliferation. The article is well organized, and the message flows well throughout the manuscript. The authors describe for the first time that two PROTAC compounds (GU3341 and CC-90009) degrade the GSPT1 protein in the context of Acute Myeloid Leukemia. Importantly, treating these compounds decreased the expression of AML genetic drivers like the RUNX1::RUNX1T1 and FUS::ERG fusion transcripts and AML cellular proliferation and viability of both cell lines and patient-derived samples. The authors produced a significant and interesting manuscript that will be of high value to the scientific community.

However, it is not clear, or the authors did not show any clear evidence of the claim they made in the discussion: “Our data show that off-target degradation of GSPT1 induces a strong antileukemic effect and that GSPT1 loss is linked to impaired expression of the RUNX1::RUNX1T1 and FUS::ERG fusion genes.” – lines 381-383. From what the authors show in the result section, the downregulation of GSPT1 by the PROTACs (particularly in the case of the GU3341) could only be a side effect of the cytotoxic action of these compounds. Is it possible to downregulate GSPT1 expression (RNAi or CRISPR/Cas9) and then incubate with the PROTACS and ascertain whether the effects on viability, proliferation and gene expression still maintain?

As minor concerns:

-        I don’t agree with the statement on lines 222-223 “Nevertheless, CDK4 protein levels rose 222 again after 72 hours, despite the continued presence of the PROTAC (Figure 2C).” The expression of CDK4 at 72h with GU3341 incubation is still low as shown by the western-blot and the densitometry calculated by the authors.

-        Figure 1D – “nM PROTAC” – it should be nM compound/drug – since in that panel Palbociclib is also added (in green) and it is not a PROTAC compound.

Author Response

The manuscript is an interesting report showing that targeting the GSPT1 protein with two different PROTAC compounds decreases leukemic cellular survival and proliferation. The article is well organized, and the message flows well throughout the manuscript. The authors describe for the first time that two PROTAC compounds (GU3341 and CC-90009) degrade the GSPT1 protein in the context of Acute Myeloid Leukemia. Importantly, treating these compounds decreased the expression of AML genetic drivers like the RUNX1::RUNX1T1 and FUS::ERG fusion transcripts and AML cellular proliferation and viability of both cell lines and patient-derived samples. The authors produced a significant and interesting manuscript that will be of high value to the scientific community.

Comment 1: However, it is not clear, or the authors did not show any clear evidence of the claim they made in the discussion: “Our data show that off-target degradation of GSPT1 induces a strong antileukemic effect and that GSPT1 loss is linked to impaired expression of the RUNX1::RUNX1T1 and FUS::ERG fusion genes.” – lines 381-383. From what the authors show in the result section, the downregulation of GSPT1 by the PROTACs (particularly in the case of the GU3341) could only be a side effect of the cytotoxic action of these compounds. Is it possible to downregulate GSPT1 expression (RNAi or CRISPR/Cas9) and then incubate with the PROTACS and ascertain whether the effects on viability, proliferation and gene expression still maintained?

Response 1: We thank the reviewer for this comment. We understand the concern regarding the potential side effects of GSPT1 downregulation due to the cytotoxic effects of the PROTACs. CC-90009 is a selective PROTAC for GSPT1 degradation. As shown in Figure 4C, GSPT1 is fully degraded after 24 hours of treatment with 10 nM or higher concentrations of CC-90009, with no observed cytotoxic effect at this time point (Figure 4B). The cytotoxic effect of CC-90009 is delayed compared to GSPT1 degradation. GU3341 also induces GSPT1 knockdown with very similar kinetics. Therefore, we believe that CC-90009 treatment serves a comparable purpose as targeting GSPT1 by RNAi or CRISP/Cas9. To clarify in the text this part (line 386-389) we revised this sentence to: “Our data show that the off-target degradation of GSPT1, together with CDK6 degradation, induces a strong antileukemic effect and impairs expression of the RUNX1::RUNX1T1 and FUS::ERG fusion genes.”

Comment 2: I don’t agree with the statement on lines 222-223 “Nevertheless, CDK4 protein levels rose 222 again after 72 hours, despite the continued presence of the PROTAC (Figure 2C).” The expression of CDK4 at 72h with GU3341 incubation is still low as shown by the western-blot and the densitometry calculated by the authors.

Response 2: We thank the reviewer for highlighting this point. We agree that after 72 hrs of treatment with GU3341, the expression of CDK4 is still lower than prior to treatment, as indicated by the western blot showing a 45% reduction in protein levels. For clarity, we changed the text (line 224-225) from “Nevertheless, CDK4 protein levels rose again after 72 hours, despite the continued presence of the PROTAC” to “ Nevertheless, CDK4 protein levels partially recovered after 72 hours, despite the continued presence of the PROTAC.”

Comment 3: Figure 1D – “nM PROTAC” – it should be nM compound/drug – since in that panel Palbociclib is also added (in green) and it is not a PROTAC compound.

Response 3: Thank you for pointing this out. We agree with this comment and we changed the x-axis legend in Figure 1D from “nM PROTAC” to “nM compound”.